# TROFI: Trajectory-Ranked Offline Inverse Reinforcement Learning

**Alessandro Sestini[1], Joakim Bergdahl[1], Konrad Tollmar[1], Andrew D. Bagdanov[2], Linus Gisslén[1]**

{asestini,jbergdahl,ktollmar,lgisslen}@ea.com,
andrew.bagdanov@unifi.it

[1]**SEED - Electronic Arts (EA), Stockholm, Sweden**
[2]**Media Integration and Communication Center (MICC), Florence, Italy**

## Abstract

In offline reinforcement learning, agents are trained using only a fixed set of stored transitions derived from a source policy. However, this requires that the dataset be labeled by a reward function. In applied settings such as video game development, the availability of the reward function is not always guaranteed. This paper proposes Trajectory-Ranked OFfline Inverse reinforcement learning (TROFI), a novel approach to effectively learn a policy offline without a pre-defined reward function. TROFI first learns a reward function from human preferences, which it then uses to label the original dataset making it usable for training the policy. In contrast to other approaches, our method does not require optimal trajectories. Through experiments on the D4RL benchmark we demonstrate that TROFI consistently outperforms baselines and performs comparably to using the ground truth reward to learn policies. Additionally, we validate the efficacy of our method in a 3D game environment. Our studies of the reward model highlight the importance of the reward function in this setting: we show that to ensure the alignment of a value function to the actual future discounted reward, it is fundamental to have a well-engineered and easy-to-learn reward function.

## 1   Introduction

In recent years, the game industry has faced a growing need to create human-like and high-quality behaviors for Non-Player Characters (NPCs) in video games (Jacob et al., 2020a). Techniques such as imitation learning and inverse reinforcement learning have been applied to this problem (Pearce et al., 2023; Biré et al., 2024; Zhang et al., 2025). Modern video game development provides multiple ways for collecting datasets of human gameplay – such as from play-testing sessions or post-release player data – resulting in large volumes of gameplay transitions that can be used to train agents. However, these datasets often include a wide variety of behaviors, differing in skill level and playstyle. Naively imitating all available behaviors can lead to suboptimal performance (Pearce et al., 2023).

Offline Reinforcement Learning (ORL) has emerged as a promising avenue for policy optimization from large datasets. ORL enables learning policies from a fixed-sized dataset of previously collected experience, sourced from an arbitrary external policy which may be either optimal or sub-optimal. In contrast to online reinforcement learning, mitigating the requirement of interacting with the environment allows ORL to be efficient even in cases where data collection is expensive, slow, or challenging due to the nature of the environment. The aim of ORL is to learn a policy that outperforms the one used to collect the experience dataset (Lange et al., 2012). A key requirement for ORL is the ability to exploit the ground truth reward used to label each transition in the training

dataset (Kumar et al., 2020; Kostrikov et al., 2021a). As a consequence, ORL requires a hand-crafted and engineered reward function. However, in applied settings such as video game development, the reward function may not always be available or easy to define. In video game development, agents are expected to exhibit human-like and enjoyable behaviors that align with gameplay dynamics. This necessitates engineering a reward function that favors qualitative behaviors (Zhao et al., 2020). One possible solution is to manually engineer a reward function and label all transitions. This poses a dual challenge: on one hand, we know that engineering a good reward function, especially for qualitative behaviors, is notoriously difficult (Open AI, 2016; Jacob et al., 2020b); on the other, the massive amount of data available renders manual labelling impractical.

Inverse Reinforcement Learning (IRL) and Imitation Learning (IL) approaches, such as Behavioral Cloning (BC) (Bain & Sammut, 1995), Generative Adversarial Imitation Learning (GAIL) (Ho & Ermon, 2016), and Adversarial Inverse Reinforcement Learning (AIRL) (Fu et al., 2018) offer techniques to learn a policy from a pre-collected dataset. The aim of IRL is to learn a reward function capturing user intent that can then be used to train a new agent that mimics the expert behavior. In contrast, IL aims to directly infer a policy from demonstrations that mimics the expert behavior (Ho & Ermon, 2016; Sestini et al., 2022; 2021; Yu et al., 2022a). These approaches have achieved great success in many different online domains.

In the ORL setting we can consider utilizing IRL or IL techniques. However, their application presents two major drawbacks: firstly, most of the popular IRL and IL methods assume that the demonstrations are optimal, while in an offline setting we might not know *a priori* what the performance of the source policy is. This is especially if the reward signal is not available, e.g., a dataset of gameplay transitions from different players with varying skill levels; secondly, many state-of-the-art approaches are adversarial techniques, designed to work only in the online setting. In this paper we investigate a pure and minimalist offline method in which we train an agent using a dataset that contains non-expert and reward-free data. We propose Trajectory-Ranked OFfline Inverse reinforcement learning (TROFI), which leverages the combination of state-of-the-art inverse and offline reinforcement learning algorithms called Trajectory-ranked Reward EXtrapolation (T-REX) (Brown et al., 2019) and Twin Delayed Deep Deterministic plus Behavioral Cloning (TD3+BC) (Fujimoto & Gu, 2021). TROFI first learns a reward model using human preferences and then automatically labels all data in the offline dataset with the learned reward model. Finally, it optimizes the agent with the newly labeled data.

The contributions of this paper are: (1) we propose a weakly-supervised method based on human preferences to learn an offline policy without the need for a reward function or optimal expert demonstrations; (2) we evaluate our approach on a set of environments and offline datasets from a 3D game environment (Sestini et al., 2023) and from the D4RL benchmark (Fu et al., 2020); (3) our approach demonstrates how game developers can efficiently leverage large-scale player data without the need of reward design; (4) we perform an empirical reward analysis by evaluating the performance achieved by a policy trained with our method as well as baselines; and (5) we show promising results that not only demonstrate that our approach surpasses state-of-the-art inverse reinforcement learning and imitation learning methods, but also the importance of having a well-defined and easily learnable reward function for offline reinforcement learning, compared to the online setting, particularly in the presence of sub-optimal data.

## 2 Preliminaries and Methodology

In this section we first introduce some preliminary concepts and terminology and then describe our approach and all of its components. A comprehensive description of the related work most relevant to our study is provided in Appendix A.

## 2.1 Problem Setting

We consider the fully observable Markov Decision Problem (MDP) setting $(S, A, R, p, \gamma)$, where $S$ is the state space, $A$ is the action space, $R$ is the reward function, $p$ is the transition probability function, and $\gamma$ is the discount factor (Sutton et al., 1998). The aim of Reinforcement Learning (RL) is to find a policy $\pi$ that maximizes the expected discounted reward $\mathbb{E}_\pi[\sum_{t=0}^\infty \gamma^t r_{t+1}]$. We measure this objective with a value function, which measures the expected discounted reward after taking action $a$ in state $s$: $Q^\pi(s, a) = \mathbb{E}_\pi[\sum_{t=0}^\infty \gamma^t r_{t+1} | s_0 = s, a_0 = a]$.

In the ORL setting we have access to a static offline dataset consisting of tuples $D = \{(s_i, a_i, s_i')\}_{i=0}^n$, where $s_i'$ is the next state. Note that in our case, we do not have rewards $r_i$. In practice, $D$ includes diverse trajectories produced by an arbitrary source policy $\pi_s$. We are interested in learning an offline policy $\pi$ from these unlabeled experiences without any other interactions with the environment. To solve this problem we introduce Trajectory-Ranked OFfline Inverse reinforcement learning (TROFI). consists of the following steps: (1) learning a reward model $\hat{r}_\theta$ with weak supervision and human preferences through T-REX; (2) labeling the dataset $D$ using the learned $\hat{r}_\theta$; and (3) training a parametrized policy $\pi_\theta$ with TD3+BC on the newly labeled dataset $\hat{D}$.

## 2.2 Reward Learning with T-REX

T-REX is an IRL approach that utilizes ranked demonstrations to extrapolate underlying user intent beyond the best demonstration (Brown et al., 2019). Specifically, it uses ranked demonstrations to learn a state-based reward function that assigns a higher reward to higher-ranked trajectories. Given a dataset $D$ of transitions, we extract a subset of $M \subset D$ ranked trajectories $\tau_t$, where $\tau_i \prec \tau_j$ if $i \prec j$, and we wish to find a parameterized reward function $\hat{r}_\theta$ that approximates the true reward function $r$. We only assume access to a qualitative ranking over a subset of trajectories. Given the ranked demonstrations, T-REX performs reward inference by approximating the reward at state $s$ using a neural network, $\hat{r}_\theta(s)$, such that $\sum_{s \in \tau_i} \hat{r}_\theta(s) < \sum_{s \in \tau_j} \hat{r}_\theta(s)$ when $\tau_i \prec \tau_j$. The reward model $\hat{r}_\theta$ can be trained using the loss:

$$\mathcal{L}(\theta) = -\sum_{\tau_i \prec \tau_j} \log \frac{\exp\sum_{s \in \tau_j} \hat{r}_\theta(s)}{\exp\sum_{s \in \tau_i} \hat{r}_\theta(s) + \exp\sum_{s \in \tau_j} \hat{r}_\theta(s)}. \tag{1}$$

Following the original paper, we train the reward model on partial trajectory pairs rather than full ones. During training we randomly select pairs of trajectories $\tau_i$ and $\tau_j$. We then select multiple partial trajectories $\hat{\tau}_i$ and $\hat{\tau}_j$ of length $L$ starting from a random timestep, and use these partial trajectories as input to our model. We compute the loss and update the model using the partial trajectories. Following the idea of Fujimoto & Gu (2021), we normalize the features of every state in the provided dataset. Given the learned reward function, we then label the entire offline dataset with $\hat{r}(s_i)$ for each $s_i \in D$.

## 2.3 Policy Learning with TD3+BC

After the reward learning and labeling steps, we have a new dataset $\hat{D} = \{s_i, a_i, s_i', \hat{r}_\theta(s_i)\}_{i=0}^n$. TROFI now seeks to optimize the policy $\pi$ using ORL. We use TD3+BC (Fujimoto & Gu, 2021), which builds upon on the online RL algorithm TD3 (Fujimoto et al., 2018). Firstly, a behavioral cloning regularization is added to the standard policy update, resulting in a policy of the form:

$$\pi = \arg\max_\pi \mathbb{E}_{(s,a) \sim D} \left[ \lambda Q(s, \pi(s)) - (\pi(s) - a)^2 \right], \tag{2}$$

where $\lambda$ is a normalization term defined as:

$$\lambda = \frac{\alpha}{\frac{1}{N} \sum_{(s_i, a_i)} |Q(s_i, a_i)|} \tag{3}$$

| Dataset | BC | GT | CONS | Random | DWBC (Xu et al., 2022) | ORIL (Zolna et al., 2020) | OTR (Luo et al., 2023) | TROFI (ours) |
|---|---|---|---|---|---|---|---|---|
| hopper-medium-v2 | $30.0 \pm 0.5$ | $59.3 \pm 1.0$ | $36.5 \pm 12.9$ | $49.0 \pm 7.6$ | $52.2 \pm 0.9$ | $74.4 \pm 1.0$ | $69.8 \pm 13.9$ | $\mathbf{80.3 \pm 1.2}$ |
| halfcheetah-medium-v2 | $36.6 \pm 0.6$ | $48.2 \pm 0.1$ | $26.8 \pm 10.2$ | $17.0 \pm 8.6$ | $42.2 \pm 0.2$ | $\mathbf{58.9 \pm 0.9}$ | $42.7 \pm 1.1$ | $55.1 \pm 0.3$ |
| walker2d-medium-v2 | $11.4 \pm 6.3$ | $\mathbf{83.8 \pm 0.3}$ | $50.1 \pm 12.4$ | $43.9 \pm 17.6$ | $66.5 \pm 2.4$ | $82.2 \pm 12.9$ | $78.0 \pm 2.6$ | $76.8 \pm 0.4$ |
| hopper-medium-replay-v2 | $19.7 \pm 5.9$ | $64.2 \pm 7.2$ | $24.9 \pm 4.7$ | $20.7 \pm 2.3$ | $17.8 \pm 8.4$ | $69.6 \pm 6.6$ | $80.2 \pm 23.1$ | $\mathbf{86.1 \pm 2.3}$ |
| halfcheetah-medium-replay-v2 | $34.7 \pm 1.8$ | $44.2 \pm 0.1$ | $31.8 \pm 3.0$ | $1.0 \pm 0.7$ | $22.7 \pm 1.0$ | $40.9 \pm 11.5$ | $38.9 \pm 1.5$ | $\mathbf{45.3 \pm 1.5}$ |
| walker2d-medium-replay-v2 | $08.3 \pm 1.5$ | $78.1 \pm 2.5$ | $25.7 \pm 7.1$ | $12.6 \pm 7.3$ | $8.5 \pm 4.2$ | $\mathbf{83.7 \pm 1.8}$ | $67.4 \pm 20.6$ | $52.7 \pm 26.4$ |
| hopper-medium-expert-v2 | $89.6 \pm 27.6$ | $98.4 \pm 1.4$ | $63.2 \pm 15.0$ | $39.8 \pm 13.0$ | $51.1 \pm 2.0$ | $81.8 \pm 24.3$ | $98.9 \pm 19.7$ | $\mathbf{99.8 \pm 3.6}$ |
| halfcheetah-medium-expert-v2 | $67.6 \pm 13.2$ | $88.9 \pm 2.9$ | $40.0 \pm 17.6$ | $12.4 \pm 10.2$ | $42.4 \pm 0.5$ | $81.7 \pm 4.2$ | $71.6 \pm 23.1$ | $\mathbf{92.5 \pm 0.8}$ |
| walker2d-medium-expert-v2 | $12.0 \pm 5.8$ | $110.2 \pm 0.4$ | $17.3 \pm 11.3$ | $39.1 \pm 17.5$ | $72.1 \pm 1.3$ | $88.1 \pm 42.6$ | $108.8 \pm 0.8$ | $\mathbf{111.1 \pm 0.9}$ |
| hopper-expert-v2 | $\mathbf{111.5 \pm 1.3}$ | $110.2 \pm 1.1$ | $95.1 \pm 29.7$ | $14.1 \pm 8.5$ | $109.1 \pm 2.2$ | $31.7 \pm 10.5$ | - | $111.3 \pm 0.3$ |
| halfcheetah-expert-v2 | $\mathbf{105.2 \pm 1.7}$ | $96.2 \pm 0.9$ | $10.9 \pm 3.7$ | $13.5 \pm 6.7$ | $89.7 \pm 1.7$ | $19.1 \pm 5.7$ | - | $96.1 \pm 0.3$ |
| walker2d-expert-v2 | $56.0 \pm 24.9$ | $110.3 \pm 0.1$ | $14.1 \pm 8.2$ | $37.1 \pm 32.1$ | $107.7 \pm 0.4$ | $93.8 \pm 42.1$ | - | $\mathbf{110.4 \pm 0.1}$ |
| 3D Game Environment medium-expert | $33.8 \pm 0.2$ | $34.4 \pm 0.5$ | $33.1 \pm 0.8$ | $17.5 \pm 4.1$ | $32.2 \pm 1.9$ | $30.9 \pm 1.8$ | - | $\mathbf{34.5 \pm 0.2}$ |
| Total | $616.4 \pm 91.3$ | $1026.4 \pm 18.5$ | $468.1.0 \pm 138.6$ | $317.7 \pm 136.2$ | $682.0 \pm 25.2$ | $836.8 \pm 165.9$ | - | $\mathbf{1051.9 \pm 38.3}$ |

Table 1: Normalized score averaged over the final 100 evaluations and 5 seeds. We highlight the best performing method for each task, as well as for the total performance. TROFI outperforms inverse ORL methods on the majority of tasks. Surprisingly, it outperforms the GT baseline on many tasks. We provide a possible explanation for this in Section 3.4. For OTR (Luo et al., 2023) we report the results from the original paper (which do not include expert tasks).

and $\alpha$ is a hyperparameter controlling the strength of the BC regularizer. Following the original paper, we use $\alpha = 2.5$ in our experiments. Secondly, the features of every state in $D$ are normalized. The entire TROFI algorithm is summarized in Appendix C. Contrary to other IRL or offline IRL methods (Konyushkova et al., 2020), our approach is completely offline as it does not require any interaction with the user during training, except for the ranking in the initial step. Moreover, as we will see in Section 3, our method works for a variety of dataset types as it does not require optimal expert demonstrations. Any offline policy optimization method could be used; however, our experiments in Appendix E indicate that TD3+BC is the best choice for our study.

## 3 Experimental Results

Our experiments aim to evaluate TROFI and answer the following research questions: (1) Can TROFI achieve performance comparable to ORL which exploits ground-truth rewards? (2) Can TROFI outperform other offline IRL baselines? (3) How does TROFI perform when varying the number of ranked trajectories? We also perform a thorough analysis of our reward model to demonstrate that an effective and easy-to-learn reward function, in addition to a good optimization algorithm, is fundamental to outperforming the source policy $\pi_s$.

### 3.1 Environments and Datasets

We first evaluate TROFI on the OpenAI Gym MuJoCo tasks using the D4RL datasets (Fu et al., 2020) in three different environments: hopper, halfcheetah, and walker2D – the same used by Fujimoto & Gu (2021), Kostrikov et al. (2021a) and Kostrikov et al. (2021b). For each environment we consider the expert, medium-expert, medium-replay, and medium datasets. Second, we further test TROFI in a 3D game environment. The environment is the one proposed by Sestini et al. (2023). This environment is an open-world, procedurally generated city simulation where the agent's objective is to navigate toward a goal by following a relatively complex trajectory and interacting with various elements in the scene. More details about the environment, such as the action- and state-space, are provided in Appendix D. For this experiment, we collect a medium-expert dataset consisting of 200,000 transitions sampled from a replay buffer of a soft actor-critic agent (Haarnoja et al., 2018), which was trained with an engineered reward function.

For all the environments, we rank a subset $M$ of the trajectories from the original dataset $D$. Similar to the oracle used by Brown et al. (2019), we approximate human preferences through an automated ranking process using the episodic rewards provided by the dataset. This strategy was chosen for

| Dataset | TROFI | TROFI-50% | TROFI-10% | TROFI-5% |
|---|---|---|---|---|
| hopper-medium-v2 | **80.3 ± 1.2** | 69.4 ± 1.5 | 72.0 ± 1.3 | 74.0 ± 1.3 |
| halfcheetah-medium-v2 | 43.2 ± 0.0 | 48.2 ± 0.2 | 53.8 ± 0.4 | **55.1 ± 0.3** |
| walker2d-medium-v2 | **76.8 ± 0.4** | 75.0 ± 0.9 | 73.3 ± 10.5 | 70.2 ± 17.5 |
| hopper-medium-replay-v2 | 70.5 ± 15.3 | **86.1 ± 2.3** | 64.6 ± 11.7 | 72.9 ± 6.5 |
| halfcheetah-medium-replay-v2 | 39.1 ± 0.6 | **45.3 ± 1.5** | 43.4 ± 0.3 | 37.0 ± 2.8 |
| walker2d-medium-replay-v2 | **52.7 ± 26.4** | 44.4 ± 22.1 | 50.5 ± 17.0 | 40.2 ± 20.7 |
| hopper-medium-expert-v2 | 95.0 ± 2.4 | 99.2 ± 5.9 | 96.4 ± 6.4 | **99.8 ± 3.6** |
| halfcheetah-medium-expert-v2 | 82.3 ± 1.5 | **92.5 ± 0.8** | 92.4 ± 1.0 | 92.1 ± 1.0 |
| walker2d-medium-expert-v2 | 109.0 ± 0.1 | **111.1 ± 0.9** | 107.9 ± 2.2 | 108.9 ± 1.0 |
| hopper-expert-v2 | 109.4 ± 2.3 | **111.3 ± 0.3** | 110.9 ± 0.4 | 110.8 ± 0.8 |
| halfcheetah-expert-v2 | 92.6 ± 0.2 | 94.8 ± 0.5 | **96.1 ± 0.3** | 92.0 ± 0.7 |
| walker2d-expert-v2 | 109.0 ± 0.1 | 110.0 ± 0.1 | 109.9 ± 0.1 | **110.4 ± 0.1** |

Table 2: Normalized scores comparing TROFI using different numbers of ranked trajectories. TROFI is largely unaffected by the number of ranked trajectories. Surprisingly, in most cases utilizing the entire dataset does not prove to be the best option. However, the reason why using a smaller number of ranked trajectories works better than using the full one remains unexplained, but we leave this question for future work.

two main reasons: it alleviates the need for extensive trajectory rankings. While we argue that the rankings generated through this process would resemble human preferences, the exploration using humans within more accessible environments is left for future research. In Section 3.2 we perform an ablation study by varying $|M|$.

For ORIL (Zolna et al., 2020) and DWBC (Xu et al., 2022), state-of-the-art ORL approaches that require optimal expert trajectories, we follow the same setting as Zolna et al. (2020) and extract a subset of well-performing episodes *from the expert dataset for each environment*. Note that this is a very important difference between TROFI and these methods: *TROFI uses only training data from each environment and does not require access to optimal datasets*. For OTR (Luo et al., 2023) we report the results of the original paper (which do not include evaluations on expert datasets).

## 3.2 Baselines and Ablations

We compare our approach against the following baselines. 1) **GT**, the TD3+BC algorithm with the ground truth reward; this method represents our expected upper bound. 2) **BC**, behavioral cloning on all of the data; this method heavily depends on the quality of the trajectories in the dataset. 3) **CONS**, the TD3+BC algorithm replacing all rewards in the dataset with a constant reward $c = 0$. 4) **Random**, the TD3+BC algorithm replacing all rewards in the dataset with a random reward $\sim \mathcal{U}(-1, 1)$. Moreover, we compare TROFI against a list of state-of-the-art approaches including: **ORIL (Zolna et al., 2020)**, **DWBC (Xu et al., 2022)**, and **OTR (Luo et al., 2023)**.

In the case of TROFI, we consider several ablations that use different percentages of ranked trajectories. We define **TROFI-X%**, where X% represents the percentage of trajectories used in dataset $D$ as ranked dataset $M$. We experiment with $X = \{100, 50, 10, 5\}$, where $X = 100$ indicates the usage of the entire dataset $D$. For cases where $X < 100$, we randomly sample trajectories from the training dataset prior to ranking them. We emphasize that $M$ is just the number of ranked trajectories used for estimating the reward, while for training the policy we use the entire $D$.

Like previous works (Fujimoto & Gu, 2021; Kostrikov et al., 2021a; Zolna et al., 2020), for all algorithms and tasks, we repeat every experiment with 5 random seeds and report the mean and standard deviation of the normalized performance of the last 100 episodes of evaluation. Additional information regarding the experiments, such as implementation details, results on the Adroit tasks and on using different optimization algorithms can be found in Appendix B and E.

| | Performance | | Average PC orig. dataset | | Average PC exp. dataset | | Average G exp. dataset | |
|---|---|---|---|---|---|---|---|---|
| | TR | GT | TR | GT | TR | GT | TR | GT |
| halfcheetah-medium-v2 | **55.1** | 48.2 | **0.56** | 0.14 | **0.21** | 0.14 | **0.18** | 0.09 |
| hopper-medium-v2 | **80.3** | 59.3 | **0.99** | 0.98 | **0.79** | 0.69 | **0.26** | 0.10 |
| halfcheetah-medium-replay-v2 | 45.3 | 44.2 | 0.59 | **0.74** | 0.48 | 0.47 | 0.14 | 0.12 |
| walker2d-medium-v2 | 76.8 | **83.8** | 0.45 | **0.53** | 0.19 | **0.36** | 0.00 | **0.10** |

Table 3: Analysis of the Performance, Pearson Correlation (PC), and Goodness (G) of the value function in selected cases. In each column we see the difference in values between TROFI (TR) and the GT baseline. *orig. dataset* means the original dataset on which the algorithm is trained, while *exp. dataset* refers to the optimal one (i.e. expert one). For each metric, the higher the better. In orange, the cases where TROFI clearly outperforms GT. In blue, the cases where GT clearly outperforms TROFI. In cases where TROFI outperforms the GT baseline, the value function is more correlated with the TROFI reward than the GT reward, and vice-versa.

## 3.3 Offline Reinforcement Learning Performance

Table 1 summarizes TROFI performance with respect to baselines and the state-of-the-art. TROFI performs comparably to the GT method on most MuJoCo tasks. In some cases it even outperforms the GT upper bound. This interesting finding is further analyzed in Section 3.4. As the table shows, ORIL performs well on some datasets, but suffers on the dataset with the most optimal data. This is due to the discriminator failing to distinguish between good and bad trajectories given a dataset with low variance in episodic reward. In contrast, our approach performs well on average on all datasets, outperforming the state-of-the-art baseline ORIL on 9 out of 12 datasets. In Appendix E we provide experiments using a variety of policy optimization algorithms. Surprisingly, for a subset of datasets, TD3+BC achieves competitive performances even with random and constant reward. This finding resembles the one found by Li et al. (2023b).

Similar to its performance in the MuJoCo environment, TROFI performs comparably, if not better, to the GT baseline in the 3D game environment. This experiment demonstrates that a complex engineered reward function can be effectively replaced by human ranking, especially when a large dataset of gameplay transitions is available. Game designers and developers can rank only a few trajectories, and so, enable the training of a high-performing agent without the need to define a traditional reward function.

Table 2 illustrates the performance of TROFI when varying the number of ranked demonstrations used to learn the reward model $\hat{r}$. As seen in the table, the decline in performance when reducing the number of ranked trajectories is generally low, even when only $5\%$ of the total dataset is ranked. Notably, using the entire dataset is not always the best choice. This increases the usability of TROFI, making it a minimalist and easy-to-use algorithm for ORL without reward functions. For example, $5\%$ of the hopper dataset is equivalent to only 10 ranked trajectories. As mentioned in Section 3.1, the results in Table 1 rely on automatic ranking based on ground-truth reward. To simulate human-generated ranking, we apply noise to the ranking for the halfcheetah medium-expert dataset and the 3D game environment dataset, thereby introducing imprecision. The detailed results of this experiment are described in Appendix E.

## 3.4 Reward Analysis

As Section 3.3 shows, on certain tasks TROFI not only outperforms the baselines, but also the GT method. This is a surprising result and we investigate it further here. Most ORL optimization algorithms are value function-based algorithms (Fujimoto & Gu, 2021; Kumar et al., 2020; Kostrikov et al., 2021a) and the general aim is to learn a function $Q_\phi(s, a)$ that represents the corresponding policy and generalizes to actions outside the training dataset $D$. This is important because out-of-distribution actions can produce erroneous values leading to a bad policy, while the value func-

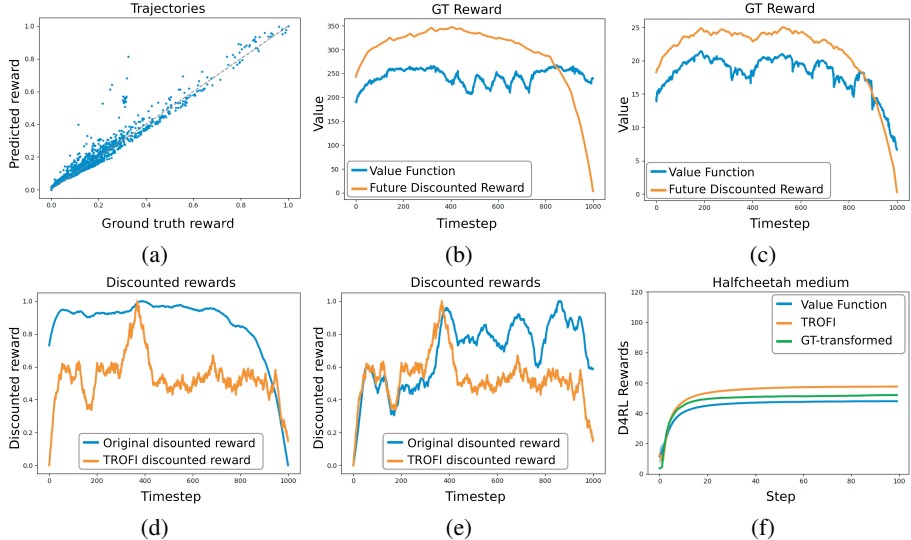

Figure 1: (a) shows the correlation between TROFI and GT reward, while (b) and (c) show the difference for one trajectory between the estimated value function – by GT first and TROFI in the last plot – and the real discounted reward used to label the dataset. The value function estimated by our approach is closer to the discounted reward compared to the GT baseline, even though the reward model estimated by TROFI is highly correlated with the GT reward. (d) example of different discounted rewards computed using TROFI and GT rewards for the same trajectory; (e) example of different discounted rewards computed using TROFI and GT rewards respectively *after the transformations* for the same trajectory; (f) performance with the transformed reward. The plots show how with simple linear modifications, we can have a discounted reward that is easier to learn for the optimization algorithm thus improving performance. Further details are provided in Section 3.4.

tion must output high values for optimal actions not seen in the dataset (Kostrikov et al., 2021a;b). $Q_\phi(s, a)$ is generally trained to minimize the temporal difference error:

$$\mathcal{L}(\phi) = \mathbb{E}_{(s,a,s')\sim D}\left[(r + \gamma \max_{a'} Q_{\hat{\phi}}(s', a') - Q_\phi(s, a))^2\right],\qquad(4)$$

where $\gamma$ is the discount factor, $Q_\phi$ is the parametric value function, and $Q_{\hat{\phi}}$ is the target value function. Note that the learning of $Q_\phi$ is highly dependent on the underlying reward function in the dataset. We investigate four cases: halfcheetah-medium and hopper-medium in which TROFI outperforms GT, walker2D-medium in which GT outperforms TROFI, and halfcheetah-medium-replay in which TROFI and GT have the same performance. Figure 1 shows an example where T-REX finds a reward function that explains the ranking of the trajectories in $M$. As GT should be the optimal reward function, and the one that TROFI learns should match it, the question is: *Why does TROFI outperform GT?*

Our explanation is based on the correlation between $Q_\phi(s, a)$, an estimator of the future discounted reward given by the underlying reward function used to label the dataset, and the actual discounted reward. To analyze this, we compute the average Pearson Correlation (PC) (Freedman et al., 2007) between $Q_\phi(s, a)$ and the discounted reward computed from the reward function used to label the offline dataset (e.g. GT or $\hat{r}$). This choice is motivated by the fact that it is not important for $Q_\phi(s, a)$ to estimate the *exact* reward value, but rather that it exhibits the same trends. Moreover, we compute the average Goodness (G) of the value function which, defined as:

$$G(Q_\phi, \tau_j) = \mathbb{E}_{(s,a^*)\sim\tau_j}\left[\frac{1}{K}\sum_{i=0}^{K-1}\mathbb{1}(Q_\phi(s, a_i) < Q_\phi(s, a^*))\right],\qquad(5)$$

where $\tau_j \in D_E$ is a trajectory from an expert dataset, $a^*$ is the optimal action for the state $s$, $a_i \in \{a_i \mid a_i \sim \mathcal{U}(A \setminus \{a^*\}), \ i = 0, .., K - 1\}$, and $\mathbb{1}(\cdot)$ returns 1 when its argument is true and 0 otherwise. With $G$ we quantify the frequency at which the optimal action has a higher value than $K$ random actions excluding the optimal one, thus providing insight into the generalization of the value function. In order to use optimal actions, we calculate $G$ exclusively for the expert dataset.

Table 3 shows the results for the four cases. Through this analysis, we see that in the cases where TROFI outperforms GT, our model has learned a better value function – one that is more correlated with the underlying future discounted reward computed from the reward function used to label the dataset and that generalizes better to out-of-distribution actions. Figure 1(a), (b) and (c) further support this analysis with qualitative results. As illustrated by the figures, it is clear that TD3+BC struggles to estimate the optimal value function with the GT reward, whereas with the TROFI reward the learned value function is more closely aligned with the discounted reward. We stress that the only thing that is changed between these two agents is the reward function. In the case where the GT outperforms our approach, the model learns a value function that is less correlated with the discounted reward. In the last case, we have similar results.

This analysis highlights how important the reward function is in ORL. In the online RL setting, the ever-changing dataset helps the value function to be better aligned with the future discounted reward and to generalize better as it experiences a diverse distribution of actions. Instead, in the offline case the action distribution representation is fixed and limited, hence it is fundamental to have a well-defined reward function that also makes the value function easy to learn. To further demonstrate this, we report on one last experiment in the halfcheetah-medium dataset. In this setup, we use the ground truth reward function but we apply linear transformations – such as scaling by a constant factor, chosen doing some preliminar experiments – to make it more similar to the reward function learned by TROFI. Figure 1(d) compares the original reward with the TROFI-generated reward, while Figure 1(e) shows the difference after applying the transformations. According to Ng et al. (1999), linear transformations of a dense reward function should preserve the optimal policy. We then train an agent using the transformed reward, and we observe improved performance (Figure 1(f)). These results suggest that the transformed reward helps improving value function approximation, enabling the policy to more accurately estimate the expected discounted reward, utlimately increasing the overall performance.

## 4 Conclusions and Limitations

In this paper, we introduced Trajectory-Ranked OFfline Inverse reinforcement learning (TROFI), an offline reinforcement learning method that trains a policy without having access to the true reward function. TROFI effectively reduces the requirements for leveraging large offline data for training agents by removing the need of engineering a reward function. Users need only to rank a limited number of trajectory samples to indicate preferred behaviors, significantly lowering the supervision burden. Our method outperforms baselines in the majority of datasets, and our qualitative comparisons show that having an easy-to-learn reward function allows the policy optimization algorithm to learn a value function better aligned with the discounted reward computed from the reward function used to label the offline dataset.

The findings presented in this work should be considered as initial, with further investigative work necessary. Future studies should extend the evaluation of TROFI to more environments to verify the generalizability of our results, particularly in relation to the learning of the value function. Moreover, the reason why using a small number of ranked trajectories in some tasks works better for TROFI than using the full dataset remains unexplained. Our qualitative comparisons show that having a good and easy-to-learn reward function allows the policy optimization algorithm to learn a value function better aligned with the discounted reward computed from the reward function used to label the offline dataset. We believe this is an under-explored challenge in offline reinforcement learning, and through this work we hope to motivate researchers to further consider this issue.

# A    Related Work

Here we review work from the recent literature most relevant to our contributions.

**Offline Reinforcement Learning.**   In ORL settings, also referred to as batch RL, we optimize a policy without relying on interactions with the environment but rather using a fixed dataset derived from a source policy (Lange et al., 2012). Many algorithmic variants have been proposed in recent years including, among others, Fisher Behavior Regularized Critic (Fisher-BRC) (Kostrikov et al., 2021a), Implicit Q-Learning (IQL) (Kostrikov et al., 2021b), and Twin Delayed Deep Deterministic plus Behavioral Cloning (TD3+BC) (Fujimoto & Gu, 2021). In general, ORL methods have two aims: train a policy that yields higher rewards than those stored in the dataset, while trying not to deviate from the behavior of the demonstrator. To do so, a significant portion of recently proposed ORL methods are based on either constrained or regularized approximate dynamic programming that regularizes the value or the action-value function (Kostrikov et al., 2021b;a; Peng et al., 2019). In this work we use TD3+BC mainly for its simplicity and good performance. Typical ORL settings assumes that the offline dataset is annotated with rewards for every transition, while in applied settings this may not always true. Thus, we require some way to learn a policy from offline datasets without this reward requirement.

**Learning without Rewards.**   A way to learn from offline datasets without rewards is to mimic the source policy with Behavioral Cloning (BC) (Bain & Sammut, 1995). Such approaches, will never learn to outperform the offline demonstrations from which the behavior is cloned. Imitation Learning (IL) and Inverse Reinforcement Learning (IRL) provide an alternative to BC. However, recent state-of-the-art approaches are mostly adversarial and are suitable only in the online reinforcement learning setting (Ho & Ermon, 2016; Fu et al., 2018). Moreover, they always require expert demonstrations. In a realistic offline setting without a reward function, we cannot know the performance of the source policy, and thus there is no easy way to label the dataset. In addition, we cannot provide new demonstrations as ORL supposes that we do not have access to the environment. Trajectory-ranked Reward EXtrapolation (T-REX) (Brown et al., 2019) is a preference-based IRL approach capable of learning a reward function from sub-optimal data. As we describe in Section 3, it is especially suitable for offline settings. Other recent notable approaches train a policy offline with just a few labeled trajectories. However, in our work we assume we have no pre-labeled trajectories (Li et al., 2023a; Hu et al., 2023; Yu et al., 2022b).

Recently, a growing body of research has focused on learning policies offline based on human preferences, without relying on reward models (Kang et al., 2023; Hejna & Sadigh, 2023; An et al., 2023). Similar to TROFI, these approaches train policies entirely offline using human preferences, but do not require the creation of a reward function. In specific contexts like game development, however, developing a reward function alongside the policy can offer advantages. During production, games frequently undergo daily changes, requiring retraining new agents. Having a pre-trained reward model helps the training and evaluation of these agents throughout all phases of game development.

**Inverse and Imitation ORL.**   More recently, a few inverse and imitation learning approaches have been proposed for ORL. Notable examples are, among others: Offline Reinforced Imitation Learning (ORIL) (Zolna et al., 2020), which, similarly to our approach, learns a reward function prior to training with the Generative Adversarial Inverse Reinforcement Learning loss (Ho & Ermon, 2016), with additional regularizations; Discriminator-Weighted Behavioral Cloning (DWBC) (Xu et al., 2022), which is an end-to-end approach to learn an effective offline policy with an augmented BC objective; and Optimal Transport Reward (OTR) (Luo et al., 2023) that learns a reward function prior to agent optimization. All of these approaches require *optimal expert demonstrations* – that is, demonstrations generated by an optimal policy. Other methods, such as Soft Q Imitation Learning (SQIL) (Konyushkova et al., 2020), achieves similar performance to ORIL but without optimal expert demonstrations. Our work differs from these approaches as it combines all their benefits: it does not require optimal expert demonstrations, it can be combined with any offline agent optimization algorithm, and it achieves good performance on a wide variety of offline datasets. Moreover, we

perform a thorough reward study that highlights the importance of having a clearly defined reward function or reward model in ORL.

## B    Implementation Details

We implemented TROFI using the PyTorch framework. The code is open-source and publicly accessible.[1] We implemented the TD3+BC algorithm from scratch following the original paper and used the original open source implementations for both ORIL and DWBC. For the experiments in Section 3.3, we used the best number of $|M|$ determined in preliminary experiments for both TROFI and baselines. In Section 3.3 we further study the effect of varying the number $|M|$ for TROFI. All training was performed on an NVIDIA RTX 2080 SUPER GPU, 8GB VRAM, a AMD Ryzen 7 3700X 8-Core CPU and 32GB of system memory.

## C    Algorithm

Algorithm 1 summarizes the entire TROFI algorithm, with a detailed description in Section 2.

---

**Algorithm 1** Training with TROFI

---

**Input:** unlabeled data $D$, number of sub-trajectories $N$, sub-trajectory length $L$
**Output:** a trained policy $\pi$ and a reward model $\hat{r}_\theta$

$\pi \leftarrow$ initialize policy
$\hat{r}_\theta \leftarrow$ initialize reward model
Normalize $s_i \in D, i = 0, .., |D|$
Sample $M \sim D$
Rank trajectories in $M$, from best to worst

**while** not converged **do**                                    ▷ **Training with T-REX**
    Sample trajectories $\tau_i, \tau_j \sim M, \tau_j \prec \tau_i$
    Sample sub-trajectories $\hat{\tau}_i^n \sim \tau_i$ and $\hat{\tau}_j^n \sim \tau_j, n = 0, ..., N-1$
    Update $\hat{r}_\theta$ using Equation 1 with $\hat{\tau}_i^n$ and $\hat{\tau}_j^n$
**end while**

Label $D$ with $\hat{r}_\theta$                                   ▷ **Label transitions with the trained reward model**

**while** not converged **do**                                    ▷ **Training with TD3+BC**
    Sample a batch $B = \{(s_n, a_n, \hat{r}_{\theta n}, s'_n), \ n = 0, ..., N-1\}$ from $D$
    Update $\pi$ using Equation 2 with $B$
**end while**
**return** $\pi$ and $\hat{r}_\theta$

---

## D    Additional 3D Game Environment Details

In this section, we provide more details on the 3D game environment cited in Section 3.1. It is an open-world city simulation originally proposed by Sestini et al. (Sestini et al., 2023). In this environment, the agent has a continuous action space of size 5, consisting of: two actions representing the relative target position, one action for strafing left and right, one action for shooting and one action for jumping. Each action is normalized between $[-1, 1]$, while the latter two are discretized in the game. The state space consists of a game goal position, represented as the $\mathbb{R}^2$ projections of the agent-to-goal vector onto the $XY$ and $XZ$ planes, normalized by the gameplay area size, along with game-specific state observations such as the agent's climbing status, contact with the ground, presence in an elevator, jump cooldown, and weapon magazine status. Observations include a list of entities and game objects that the agent should be aware of, e.g., intermediate goals, dynamic objects, enemies, and other assets that could be useful for achieving the final goal. For these entities, the same relative information from agent-to-goal is referenced, except as agent-to-entity. Additionally, a 3D semantic map is used for local perception. This map is a categorical discretization of the space and elements around the agent. Each voxel in the map carries a semantic integer value describing the type of object at the corresponding game world position. For this work, we use a semantic

---

[1]Repository will be published upon acceptance.

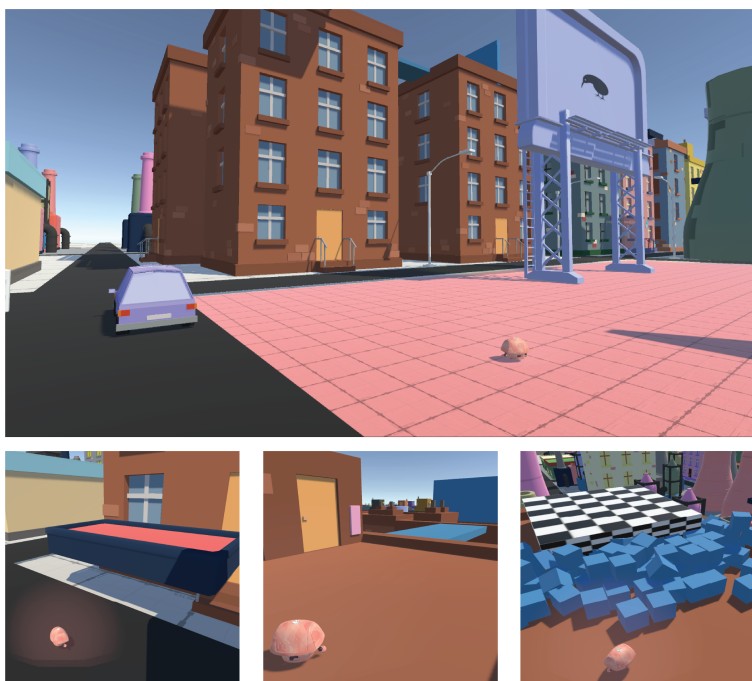

Figure 2: **Example of a trajectory in *Task 2***. The agent's starting position is on the ground, and it has to navigate to a elevator, wait for it to come down and jump over it. Once it is up on the building, the agent needs to cross a bridge between two buildings: if it falls, there is no way to get back on track. The agent has to shoot at a destructible wall in order to reveal the goal location. This example is showing the TROFI policy acting in the environment.

map of size $5 \times 5 \times 5$. In this environment, an episode consists of a maximum of 1000 steps. An episode is marked a success if the agent reaches the goal before the timeout. The environment is particularly meaningful for this study because it uses standard state- and action-spaces for developing RL agents for in-game NPCs. In fact, it is common to avoid using image-based agents, as they are too expensive at runtime, and instead use floating-point information gathered from the game engine as the state space, similar to traditional scripted game-AI. While there are no specific preferences between discrete or continuous action-spaces, it is common to use the latter in this domain (Wurman et al., 2022; Gillberg et al., 2023). Figure 2 shows an agent trained with TROFI solving the task in the environment.

## E    Additional Experimental Results

In Table 4 we compare TROFI with baselines and state-of-the-art approaches described in Section 3.2 of the main paper in two different sets of tasks: MuJoCo and Adroit. We show the training method in Section 4.1 of the main paper. While TROFI has the highest aggregate performance for MuJoCo tasks, all offline reinforcement learning algorithms struggle with most of the Adroit tasks, and in fact the behavioral cloning baseline outperforms all the others.

Table 5 summarizes the performance of different agent optimization algorithms for the same set of tasks: MuJoCo and Adroit. We compare the TD3+BC with two other populuar optimization algorithms: IQL (Kostrikov et al., 2021b) and Fisher-BRC (Kostrikov et al., 2021a). TD3+BC outperforms the other methods in the MuJoCo datasets, while IQL is better in the Adroit tasks. However, the behavioral cloning baseline still outperforms IQL in these environments.

Table 6 summarizes the performance of TROFI, in particular the 10% version of our approach (using only 10% of the entire dataset as ranked trajectories), simulating human-generated ranking. For this

experiment, we first rank the trajectories based on the ground truth reward. Then, we randomly swap the position of 20% of all the ranked trajectories. The table show that TROFI reaches high and stable performance even in case where the ranking is not optimal.

| Dataset | BC | GT | CONS | Random | DWBC (Xu et al., 2022) | ORIL (Zolna et al., 2020) | OTR (Luo et al., 2023) | TROFI (ours) |
|---|---|---|---|---|---|---|---|---|
| hopper-medium-v2 | 30.0 ± 0.5 | 59.3 ± 1.0 | 36.5 ± 12.9 | 49.0 ± 7.6 | 52.2 ± 0.9 | 74.4 ± 1.0 | 69.8 ± 13.9 | **80.3 ± 1.2** |
| halfcheetah-medium-v2 | 36.6 ± 0.6 | 48.2 ± 0.1 | 26.8 ± 10.2 | 17.0 ± 8.6 | 42.2 ± 0.2 | **58.9 ± 0.9** | 42.7 ± 1.1 | 55.1 ± 0.3 |
| walker2d-medium-v2 | 11.4 ± 6.3 | **83.8 ± 0.3** | 50.1 ± 12.4 | 43.9 ± 17.6 | 66.5 ± 2.4 | 82.2 ± 12.9 | 78.0 ± 2.6 | 76.8 ± 0.4 |
| hopper-medium-replay-v2 | 19.7 ± 5.9 | 64.2 ± 7.2 | 24.9 ± 4.7 | 20.7 ± 2.3 | 17.8 ± 8.4 | 69.6 ± 6.6 | 80.2 ± 23.1 | **86.1 ± 2.3** |
| halfcheetah-medium-replay-v2 | 34.7 ± 1.8 | 44.2 ± 0.1 | 31.8 ± 3.0 | 1.0 ± 0.7 | 22.7 ± 1.0 | 40.9 ± 11.5 | 38.9 ± 1.5 | **45.3 ± 1.5** |
| walker2d-medium-replay-v2 | 08.3 ± 1.5 | 78.1 ± 2.5 | 25.7 ± 7.1 | 12.6 ± 7.3 | 8.5 ± 4.2 | **83.7 ± 1.8** | 67.4 ± 20.6 | 52.7 ± 26.4 |
| hopper-medium-expert-v2 | 89.6 ± 27.6 | 98.4 ± 1.4 | 63.2 ± 15.0 | 39.8 ± 13.0 | 51.1 ± 2.0 | 81.8 ± 24.3 | 98.9 ± 19.7 | **99.8 ± 3.6** |
| halfcheetah-medium-expert-v2 | 67.6 ± 13.2 | 88.9 ± 2.9 | 40.0 ± 17.6 | 12.4 ± 10.2 | 42.4 ± 0.5 | 81.7 ± 4.2 | 71.6 ± 23.1 | **92.5 ± 0.8** |
| walker2d-medium-expert-v2 | 12.0 ± 5.8 | 110.2 ± 0.4 | 17.3 ± 11.3 | 39.1 ± 17.5 | 72.1 ± 1.3 | 88.1 ± 42.6 | 108.8 ± 0.8 | **111.1 ± 0.9** |
| hopper-expert-v2 | **111.5 ± 1.3** | 110.2 ± 1.1 | 95.1 ± 29.7 | 14.1 ± 8.5 | 109.1 ± 2.2 | 31.7 ± 10.5 | - | 111.3 ± 0.3 |
| halfcheetah-expert-v2 | **105.2 ± 1.7** | 96.2 ± 0.9 | 10.9 ± 3.7 | 13.5 ± 6.7 | 89.7 ± 1.7 | 19.1 ± 5.7 | - | 96.1 ± 0.3 |
| walker2d-expert-v2 | 56.0 ± 24.9 | 110.3 ± 0.1 | 14.1 ± 8.2 | 37.1 ± 32.1 | 107.7 ± 0.4 | 93.8 ± 42.1 | - | **110.4 ± 0.1** |
| Total Mujoco | 582.6± 91.1 | 992.0± 18.0 | 436.0± 137.8 | 300.2± 132.1 | 682.0± 25.2 | 805.9± 164.1 | - | **1017.4± 38.1** |
| 3D Game Environment medium-expert | 33.8 ± 0.2 | 34.4 ± 0.5 | 33.1 ± 0.8 | 17.5 ± 4.1 | 00.0 ± 0.0 | 30.9 ± 1.8 | - | **34.5 ± 0.2** |
| pen-human-v1 | **99.7 ± 7.4** | – 2.5 ± 0.5 | – 2.2 ± 1.1 | – 3.0 ± 0.5 | 37.5 ± 15.1 | – 2.7 ± 0.3 | 66.8 ± 21.2 | – 2.8 ± 0.7 |
| door-human-v1 | **9.4 ± 4.5** | – 0.3 ± 0.0 | – 0.3 ± 0.0 | – 0.3 ± 0.0 | 2.5 ± 2.2 | – 0.3 ± 0.0 | 5.7 ± 2.7 | – 0.3 ± 0.0 |
| hammer-human-v1 | **12.6 ± 4.9** | 1.0 ± 0.1 | 1.0 ± 0.1 | 1.3 ± 0.2 | 1.1 ± 0.4 | 1.0 ± 0.4 | 1.8 ± 1.4 | 1.1 ± 0.2 |
| relocate-human-v1 | **0.6 ± 0.3** | – 0.3 ± 0.0 | – 0.3 ± 0.0 | – 0.3 ± 0.0 | 1.0 ± 0.0 | – 0.3 ± 0.0 | 0.1 ± 0.1 | – 0.3 ± 0.0 |
| pen-cloned-v1 | **99.1 ± 12.3** | 10.2 ± 8.3 | 6.5 ± 8.0 | 6.4 ± 8.6 | 25.5 ± 11.8 | 11.2 ± 9.1 | 46.8 ± 20.8 | 11.1 ± 7.1 |
| door-cloned-v1 | **3.4 ± 0.9** | – 0.3 ± 0.0 | – 0.3 ± 0.0 | – 0.3 ± 0.0 | – 0.1 ± 0.0 | – 0.3 ± 0.0 | 0.0 ± 0.0 | – 0.3 ± 0.0 |
| hammer-cloned-v1 | **8.9 ± 4.0** | 0.3 ± 0.0 | 0.2 ± 0.0 | 0.2 ± 0.0 | 0.2 ± 0.0 | 0.3 ± 0.0 | 6.5 ± 8.0 | 0.2 ± 0.0 |
| relocate-cloned-v1 | **0.4 ± 0.3** | – 0.3 ± 0.0 | – 0.3 ± 0.0 | – 0.3 ± 0.0 | – 0.1 ± 0.0 | – 0.3 ± 0.0 | – 0.2 ± 0.0 | – 0.3 ± 0.0 |
| pen-expert-v1 | **128.8 ± 5.9** | 114.9 ± 19.7 | 14.5 ± 11.3 | 31.9 ± 10.5 | 76.1 ± 11.4 | 22.0 ± 8.0 | - | 108.1 ± 9.0 |
| door-expert-v1 | **105.8 ± 0.2** | – 0.3 ± 0.0 | – 0.3 ± 0.0 | – 0.3 ± 0.0 | 0.8 ± 0.6 | – 0.3 ± 0.0 | - | – 0.3 ± 0.0 |
| hammer-expert-v1 | **127.9 ± 0.5** | 3.0 ± 0.3 | 4.4 ± 2.5 | 3.6 ± 2.7 | 22.4 ± 18.5 | 0.5 ± 0.3 | - | 2.5 ± 1.7 |
| relocate-expert-v1 | **110.3 ± 0.4** | – 1.4 ± 0.2 | 0.0 ± 0.0 | – 1.1 ± 0.1 | 4.2 ± 0.2 | – 1.4 ± 1.3 | - | – 1.4 ± 0.3 |
| Total Adroit | **706.6± 41.6** | 124.0± 29.1 | 22.9± 23.0 | 37.8± 22.5 | 171.2± 60.2 | 29.4± 19.4 | - | 128.4± 19.0 |

Table 4: Normalized score averaged over the final 100 evaluations and 5 seeds. We highlight the best performing method for each task as well for the total performance across all environments and tasks. We compare our approach (TROFI) to a set of baselines – Behavioral Cloning (BC), Ground Truth (GT) reward, Constant (CONS) reward, and random reward – and state-of-the-art offline inverse reinforcement- and imitation-learning algorithms – DWBC (Xu et al., 2022) and ORIL (Zolna et al., 2020). We provide more details about the baselines in Section 4.2 of the main paper. We test our approach in two different set of tasks: MuJoCo and Adroit. TROFI outperforms both baselines and state-of-the-art algorithms for the MuJoCo tasks, while for the Adroit ones – that are hard tasks to solve with any offline reinforcement learning algorithms – the BC baseline is still the best performing one. For OTR (Luo et al., 2023) we report the results from the original paper (which do not include expert tasks).

| Dataset | IQL | Fisher-BRC | TD3+BC (ours) |
|---|---|---|---|
| hopper-medium-v2 | $20.7 \pm 4.5$ | $\mathbf{89.7 \pm 1.2}$ | $80.3 \pm 1.2$ |
| halfcheetah-medium-v2 | $42.7 \pm 0.1$ | $43.3 \pm 0.3$ | $\mathbf{55.1 \pm 0.3}$ |
| walker2d-medium-v2 | $\mathbf{79.2 \pm 1.0}$ | $75.3 \pm 2.4$ | $76.8 \pm 0.4$ |
| hopper-medium-replay-v2 | $83.9 \pm 2.2$ | $66.7 \pm 15.9$ | $\mathbf{86.1 \pm 2.3}$ |
| halfcheetah-medium-replay-v2 | $42.7 \pm 0.1$ | $42.2 \pm 0.2$ | $\mathbf{45.3 \pm 1.5}$ |
| walker2d-medium-replay-v2 | $\mathbf{73.9 \pm 1.7}$ | $57.5 \pm 23.3$ | $52.7 \pm 26.4$ |
| hopper-medium-expert-v2 | $91.5 \pm 14.3$ | $99.2 \pm 3.6$ | $\mathbf{99.8 \pm 3.6}$ |
| halfcheetah-medium-expert-v2 | $89.6 \pm 1.4$ | $\mathbf{93.2 \pm 0.8}$ | $92.5 \pm 0.8$ |
| walker2d-medium-expert-v2 | $105.3 \pm 3.8$ | $101.3 \pm 14.8$ | $\mathbf{111.1 \pm 0.9}$ |
| hopper-expert-v2 | $110.0 \pm 0.9$ | $\mathbf{111.4 \pm 0.2}$ | $111.3 \pm 0.3$ |
| halfcheetah-expert-v2 | $92.9 \pm 0.1$ | $93.6 \pm 0.5$ | $\mathbf{96.1 \pm 0.3}$ |
| walker2d-expert-v2 | $108.7 \pm 0.1$ | $108.5 \pm 0.1$ | $\mathbf{110.4 \pm 0.1}$ |
| Total MuJoCo | $941.3 \pm 30.2$ | $981.9 \pm 63.3$ | $\mathbf{1017.5 \pm 38.1}$ |
| pen-human-v1 | $\mathbf{67.3 \pm 2.6}$ | $-1.4 \pm 0.4$ | $-2.8 \pm 0.7$ |
| door-human-v1 | $-4.1 \pm 1.6$ | $\mathbf{0.2 \pm 0.3}$ | $-0.3 \pm 0.0$ |
| hammer-human-v1 | $1.0 \pm 0.6$ | $0.3 \pm 0.5$ | $\mathbf{1.1 \pm 0.2}$ |
| relocate-human-v1 | $\mathbf{0.1 \pm 0.5}$ | $0.0 \pm 0.0$ | $-0.3 \pm 0.0$ |
| pen-cloned-v1 | $\mathbf{61.7 \pm 4.0}$ | $-1.1 \pm 1.7$ | $11.1 \pm 7.1$ |
| door-cloned-v1 | $-0.5 \pm 0.0$ | $-2.0 \pm 0.0$ | $\mathbf{-0.3 \pm 0.0}$ |
| hammer-cloned-v1 | $0.2 \pm 0.3$ | $\mathbf{0.4 \pm 0.2}$ | $0.2 \pm 0.0$ |
| relocate-cloned-v1 | $\mathbf{0.0 \pm 0.0}$ | $0.0 \pm 0.0$ | $-0.3 \pm 0.0$ |
| pen-expert-v1 | $\mathbf{115.2 \pm 9.2}$ | $39.1 \pm 9.1$ | $108.1 \pm 9.0$ |
| door-expert-v1 | $\mathbf{104.9 \pm 0.2}$ | $0.2 \pm 0.3$ | $-0.3 \pm 0.0$ |
| hammer-expert-v1 | $\mathbf{127.6 \pm 0.5}$ | $2.6 \pm 4.6$ | $2.5 \pm 1.7$ |
| relocate-expert-v1 | $\mathbf{105.0 \pm 0.0}$ | $0.0 \pm 0.0$ | $-1.4 \pm 0.3$ |
| Total Adroit | $\mathbf{588.3 \pm 19.5}$ | $37.9 \pm 16.1$ | $128.4 \pm 19.0$ |

Table 5: Aggregate performance of different agent optimization methods. We replace the agent optimization approach in TROFI with three distinct offline reinforcement learning algorithms: IQL (Kostrikov et al., 2021b), Fisher-BRC (Kostrikov et al., 2021a), and TD3+BC (Fujimoto & Gu, 2021). Of the three, TD3+BC proves to be the most suitable algorithm for MuJoCo tasks, while IQL performs better than the others on the Adroit tasks but only in the most expert datasets. In this scenario, the BC baseline still outperforms all offline reinforcement learning algorithms.

| Dataset | human-generated ranking | optimal ranking |
|---|---|---|
| halfcheetah medium-expert | $90.0 \pm 2.6$ | $92.5 \pm 0.8$ |
| 3D game environment | $33.8 \pm 0.3$ | $34.5 \pm 0.2$ |

Table 6: Comparison between simulated human-generated ranking and optimal ranking. After we rank the trajectories following the ground truth reward, we apply noise into the ranking to simulate how humans would order the episodes. The table shows that, although the performance degrades slightly, TROFI is able to reach high and stable performance.

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
