# OpenReview forum: "TROFI: Trajectory-Ranked Offline Inverse Reinforcement Learning"
_rl-conference.cc/RLC/2025/Workshop/RLVG — RLVG Workshop - RLC 2025_

### Official Review · Reviewer_8E9Y · 2025-06-15
**well-witten work targeting Important use-case but lacking novelty**

**Rating:** 4
**Confidence:** 4

**Summary:**

Paper Trajectory-Ranked Offline Inverse Reinforcement Learning (TROFI) proposes a two step solution for reward-labeling of unlabeled and sub-optimal datasets to be used for policy learning in an offline reinforcement learning (RL) setting. In the first step, inverse reinforcement learning is used to learn a human-aligned reward model from the partial trajectories. The original dataset is then labeled using this reward model. Their proposed method effectively combines two state-of-the-art methods, Trajectory-ranked Reward EXtrapolation (T-REX) (Brown et al., 2019) and Twin Delayed Deep Deterministic plus Behavioral Cloning (TD3+BC) (Fujimoto & Gu, 2021).
They further provide reward analysis supported by empirical results.

**Strengths:**

- The paper provides a solution for an important real-world challenge in games use-cases, namely, the lack of labeled data for training RL agents
- TROFI performs comparable to GT, the TD3+BC algorithm with the ground truth reward
- Better value function alignment to the discounted reward  compared with the baselines
- The solution works with a sub-optimal dataset
- Provided an ablation study on the effect of the number of ranked trajectories

**Weaknesses:**

- The work provides good analysis but lacks novelty as it combines two already existing methods
- The preference ranking mechanism is a key part of the method, which is not explained properly.
- I assume the ground truth rewards in the Mujoco datasets are not that complex or engineered, so to claim that “ This experiment demonstrates that a complex engineered reward function can be effectively replaced by human ranking,” you need to show a comparison with more complex reward functions.
- Clarify on the ground truth reward fucntions on each dataset specifically the 3D env

**Best Paper Nomination:**

No

**Claims:**

Yes, the claims are supported by the provided evidence.

**Suggestions:**

In sections 2.2 and 3.1  provide details of how you collect or approximate the human preference rankings.

---

### Official Review · Reviewer_zQyf · 2025-06-16
**Interesting problem and paper, limited novelty**

**Rating:** 3
**Confidence:** 4

**Summary:**

Trajectory-Ranked OFfline Inverse reinforcement learning (TROFI) introduces an approach to learning policies from offline data without rewards by using human preferences. It involves using T-REX to learn a reward function from preferences and then TD3-BC to learn a policy using the learned reward function, therefore effectively combining two existing approaches to provide an offline preference-based RL approach.

**Strengths:**

- Considers important problem - learning an optimal policy from suboptimal offline data without reward labels.
- Well motivated for the context of games.
- Strong results, comparable to GT rewards.
- Interesting ablation on the preference requirements (which is crucial for this approach) demonstrating its effectiveness.

**Weaknesses:**

- Limited novelty - effectively T-REX in the offline setting by using TD3-BC rather than PPO.
- Incomplete aspects - some unanswered questions could do with further investigation/analysis e.g. improved performance with fewer preferences.
- Use of GT rewards to obtain preferences.
- Use of D4RL rather than video game/more complex reward environments.

**Best Paper Nomination:**

No

**Claims:**

Yes, the claims are supported.

**Suggestions:**

- Provide additional justification of the contribution over T-REX - how is TROFI different and why is that better? This could incorporate some of the additional related work in Appendix A e.g. paragraph starting at line 416.
- L150 says the strategy was chosen for two main reasons but appears to only provide one - what is the other?
- Investigating additional video game environments with more complex reward functions would strengthen the claims and novelty of the paper.

---

### Decision · Program_Chairs · 2025-06-19

**Decision:**

Accept

**Comment:**

This paper introduces Trajectory-Ranked Offline Inverse Reinforcement Learning (TROFI), a two-step approach that combines T-REX and TD3-BC to learn policies from unlabeled, suboptimal offline data in an offline reinforcement learning setting by leveraging human preferences to learn a reward function.

The paper's strengths lie in its focus on the important problem of learning from suboptimal offline data without reward labels, offering a well-motivated solution for games with strong results comparable to ground truth rewards and an insightful ablation study on preference requirements.

However, reviewers had concerns with the paper's limited novelty, as it primarily combines existing methods, and a lack of detailed explanation regarding the preference ranking mechanism. We encourage the authors to address these points for the camera-ready version by providing additional justification for TROFI's unique contributions beyond merely combining T-REX and TD3-BC, clarifying the preference ranking methodology, and extending experiments to more complex video game environments with intricate reward functions. This would help to improve the camera-ready version for presentation at the workshop.